# Update on Lean Body Mass Diagnostic Assessment in Critical Illness

**DOI:** 10.3390/diagnostics13050888

**Published:** 2023-02-26

**Authors:** Silvia De Rosa, Michele Umbrello, Paolo Pelosi, Denise Battaglini

**Affiliations:** 1Centre for Medical Sciences—CISMed, University of Trento, Via S. Maria Maddalena 1, 38122 Trento, Italy; 2Anesthesia and Intensive Care, Santa Chiara Regional Hospital, APSS, 38123 Trento, Italy; 3S.C. Anestesia e Rianimazione II, Ospedale San Carlo Borromeo, ASST dei Santi Paolo e Carlo, 20142 Milano, Italy; 4IRCCS Ospedale Policlinico San Martino, 16132 Genova, Italy; 5Dipartimento di Scienze Chirurgiche e Diagnostiche Integrate, Università degli Studi di Genova, 16132 Genova, Italy

**Keywords:** critical care, lean body mass, muscle mass, assessment, CT scan, bioelectrical impedance analysis, electromyography, musculoskeletal ultrasound

## Abstract

Acute critical illnesses can alter vital functions with profound biological, biochemical, metabolic, and functional modifications. Despite etiology, patient’s nutritional status is pivotal to guide metabolic support. The assessment of nutritional status remains complex and not completely elucidated. Loss of lean body mass is a clear marker of malnutrition; however, the question of how to investigate it still remains unanswered. Several tools have been implemented to measure lean body mass, including a computed tomography scan, ultrasound, and bioelectrical impedance analysis, although such methods unfortunately require validation. A lack of uniform bedside measurement tools could impact the nutrition outcome. Metabolic assessment, nutritional status, and nutritional risk have a pivotal role in critical care. Therefore, knowledge about the methods used to assess lean body mass in critical illnesses is increasingly required. The aim of the present review is to update the scientific evidence regarding lean body mass diagnostic assessment in critical illness to provide the diagnostic key points for metabolic and nutritional support.

## 1. Introduction

After a critical illness, survivors who have been admitted to an intensive care unit (ICU) often present with reduced physical function. This could be one of the long-term effects of acute skeletal muscle atrophy and neuromuscular weakness acquired during critical illness as a result of extended bed rest, systemic inflammation, and bioenergetic failure [1,2,3,4,5]. Intensive care-acquired weakness (ICUAW) is a ‘clinically detected weakness in critically ill patients in whom there is no plausible etiology other than critical illness’ [6]. ICUAW is a substantial contributor to long-term disability in survivors of critical illness. Patients with ICUAW are then classified according to those with critical illness polyneuropathy (CIP), critical illness myopathy (CIM), or critical illness neuromyopathy (CINM). In addition, a further subclassification (histologically) for CIM includes cachectic myopathy, thick filament myopathy, and necrotizing myopathy [7]. Despite the influence of ICUAW on functional outcome being clearly established, the literature is scarce regarding muscle mass status upon admission to ICU. Data on pre-admission muscle status in ICU patients are currently available only for certain chronic disease groups [8,9]. The quantification of pre-ICU muscle mass in larger ICU populations and the determination of whether it may be a factor in post-ICU functional impairment are key questions that have only been partially solved [10]. Lean body mass (LBM) represents non-adipose tissue mass, excluding any additional mass from sudden changes in water content [11]. Several tools, including computed tomography (CT), ultrasound (US) imaging, and bioelectrical impedance analysis (BIA) are used to measure lean body mass. Particularly, US and abdominal CT scans are new emerging tools for body composition assessment in ICU patients, although further validation of these techniques in the ICU population is still needed. However, the effectiveness of muscle mass monitoring is helpful in guiding adequate nutritional support during the acute critical illness, recovery phase, and rehabilitation periods. In addition, a lack of uniform bedside monitoring tools could impact the nutrition outcome. Metabolic assessment has a pivotal role in critical care, and understanding the patients’ nutritional status and risk is crucial [12]. Several methods used to monitor LBM are becoming increasingly used in the ICU, and knowledge about their advantages and limitations is essential.

The aim of the present review is to update the scientific evidence regarding emerging imaging techniques for LBM diagnostic assessment in critical illnesses to provide diagnostic key points for metabolic and nutritional support.

### 1.1. Computed Tomography Assessment of Nutritional Status and Lean Body Mass

The amount and quality of skeletal muscles can be assessed using cross-sectional imaging modalities such as computed tomography (CT). Increasing evidence has demonstrated the prognostic value of the area and the quality of skeletal muscle measured using CT as biomarkers of sarcopenia and frailty. Analysis of the muscle cross-sectional area (CSA) on a single cross-sectional image at the level of the third vertebra (L3) is an accurate surrogate of whole-body muscle mass [13] (Figure 1).

The determination of circumferential skeletal muscle area or psoas muscle area, both typically obtained at lumbar vertebral levels, is the most common approach. In addition, this region contains visceral, subcutaneous, and intermuscular adipose tissue, psoas and paraspinal muscles, transversus abdominus, external and internal oblique abdominals, and rectus abdominus. De Marco et al. [14] assessed abdominal CT images in a cohort of healthy patients to define the normal reference values and age-associated down-trend for CT muscle parameters at L4 in a healthy population. The lower reference range for the psoas wall muscle area was <22.0 cm^2^ in males and <11.1 cm^2^ in females, and, for the abdominal wall muscle area, it was <112.2 cm^2^ in males and <75.6 cm^2^ in females. There was a graded decline observed among older compared to younger adults (especially ≥60 years of age). Toledo et al. [15], in their observational study, found that sarcopenia was a risk factor in lower 30-day survival, higher hospital mortality, and higher complications in critically ill patients. Moreover, there was a low correlation between sarcopenia and body mass index, whereas Looijaard et al. found that low skeletal muscle quality at ICU admission, assessed using CT scan, was independently associated with higher 6-month mortality in mechanically ventilated patients [16].

In critical care, sarcopenic obesity is prevalent but scarcely investigated. Severe muscle depletion or sarcopenia is one of the most common complications of acute and chronic illnesses. In addition, the CT analysis was not biased by the fluid overload that frequently presents in critically ill patients [17]. In a critical care setting, a CT scan could be useful for analyzing sarcopenia, sarcopenic obesity, and myosteatosis using the third lumbar vertebrae skeletal muscle [18]. Despite being a relatively new measure tool in critical illnesses, in chronic diseases, the use of a CT scan to analyze sarcopenia was well investigated. Joppa et al. showed that the prevalence of sarcopenic obesity is 2.5 times higher in chronic obstructive pulmonary disease, associated with worse physical performance and higher systemic inflammatory burden [19]. Although CT is considered a gold standard method used to assess body composition [20], its use in critically ill patients is limited despite the advocation of the substantial depletion of skeletal muscle-standardized approaches for determining muscularity. Implementation of this technique could offer several advantages for the care of critically ill patients, although newer methods with easier bedside availability are emerging, as explained below.

### 1.2. Bioelectrical Impedance Analysis

In critical care, real-time knowledge of body composition (fat, muscle, bone, and water) is advantageous for personalization and clinical optimization in terms of nutrition, fluids, and medication dosing adjustment. Bioelectrical impedance analysis (BIA) is a safe, quick, and inexpensive technique for the assessment of body composition [21].

BIA analyzers inject an alternating sinusoidal electric current through active electrodes and register resistance and reactance through recording electrodes [22]. This method estimates body fat and muscle mass, whereby a weak electric current flows through the body and the voltage is measured to calculate the impedance (resistance) of the body. Based on the principle that body water is stored in muscle, a person with more muscle mass has a higher probability of having more body water, which leads to lower impedance. The BIA technique requires the operator to place active electrodes in the right side on conventional metacarpal and metatarsal lines and record electrodes in standard positions at wrist and ankle. The measuring of the phase angle (PA) or the “classic” bioelectrical impedance vector analysis (“classic” BIVA) have emerged as alternative techniques to overcome the limitations of BIA, basing their main strength on the use of raw impedance parameters [23]. Particularly, BIVA has been assessed in its valuation of the impact of hyperhydration on ICU mortality in critically ill patients [24,25,26]. However, experience with BIA- or BIVA-guided fluid management in the ICU is limited. The substantial difference lies in the fact that the BIA method processes the measurements made through software, returning estimated values of the parameters listed above. Instead, the BIVA method, in addition to processing the parameters found in equations, includes the parameters in a biavector graph.

Despite these several advantages, some limitations can influence the accuracy, including instrument-related factors (i.e., electrodes quality), technician-related factors (inter- and intra-operator variability), subject-related factors (i.e., supine position with each limb slightly away from the body, after an overnight fast, and once the bladder is emptied), and environment-related factors (i.e., environmental temperature) [27,28,29].

Loojaard et al., in a prospective observational study enrolling 110 critically ill patients, compared the bioelectrical impedance analysis of (BIA)- and CT-derived muscle mass to determine whether BIA identified the patients with a low skeletal muscle area on the CT scan and to determine the relation between the raw BIA and raw CT measurements. BIA identified critically ill patients with a low skeletal muscle area on the CT scan, as defined by previously found cutoffs, and the BIA-derived low phase angle corresponded to low CT-derived skeletal muscle area and density [30].

Nakanishi et al. [31] investigated muscle mass monitoring capacity using BIA and an ultrasound through an assessment of fluid balance. The authors found that muscle mass monitoring using BIA was complicated by the fluid shift and could not monitor the change of muscle mass in critically ill patients, although muscle mass assessment at one point moderately correlated with ultrasound and CT. In contrast, the use of ultrasound-to monitor progressive muscle atrophy was used throughout the ICU stay, without the influence of a fluid shift.

Despite a correlation being found between raw impedance parameters, fluid ratios, overhydration, and the adverse outcome of critical illness, cutoff and reference values remain elusive. BIA-derived muscle mass could be a promising biomarker for sarcopenia, correlating well with CT-analysis. Its use is still limited in critical care, however observational data are encouraging, inviting the implementation of this technique.

### 1.3. Musculoskeletal Ultrasound

Ultrasounds are increasingly being used to assess changes in muscle size and quality over time [32]. The advantages of this include the high axial resolution, low procedural risks, absence of ionizing radiation, and ease of use, even early in the course of disease. Below, we provide an update on the most common techniques used to assess muscle ultrasound in critical care.

#### 1.3.1. Respiratory Muscles Ultrasound

Various insults, including invasive mechanical ventilation, sepsis, electrolytes disbalance, critical illness polyneuropathy, and myopathy can contribute to the contractile dysfunction of respiratory muscles. Up to 40% of patients admitted to the ICU are invasively and mechanically ventilated, and the use of a controlled passive ventilator modality has been associated with a reduction in breathing and diaphragm contractility, thus causing ventilator-induced diaphragm dysfunction (VIDD) [33]. Moreover, spontaneous breathing and assisted ventilatory modalities have been associated with potential diaphragm atrophy or damage to the myofibers, depending on the stage of the disease in which they were applied (i.e., spontaneous breathing resulted in myofibers damage if allowed too early during severe lung injury) [34]. The diaphragm contributes to 60–70% of the respiratory workload. Therefore, diaphragmatic dysfunction, defined as a loss of diaphragm force-generating capacity specifically associated with the use of mechanical ventilation, is a possible determinant of respiratory failure in critically ill patients [35,36]. Diaphragmatic dysfunction is more common in patients who are ventilated for longer but can also be observed even after relatively short periods of mechanical ventilation [37]. The assessment of intercostal muscle thickness has been recently proposed to complete the evaluation of respiratory muscular dysfunction [38]. The monitoring of respiratory muscle function in ICU patients is still an uncommon practice, although recent techniques for assessing readiness for weaning from the ventilator, respiratory muscle function, and strength have been applied [39,40]. Among these techniques, ultrasound imaging has increased in popularity since it is a rapid, accurate, and repeatable tool, which allows for optimal diagnostic accuracy and is easily available at the bedside [41]. However, ICU clinicians are still poorly trained to use ultrasounds for evaluating the function of the respiratory muscles [42]. The effective use of an ultrasound for examining the respiratory muscles requires the study of the diaphragm and the accessory inspiratory (parasternal, external intercostal, scalene, and sternocleidomastoid) and expiratory muscles (transversus abdominis muscle, internal and external oblique muscle) [42]. The measuring of diaphragmatic function includes an assessment of thickness, thickening, and displacement.

##### Diaphragmatic Thickness

Diaphragmatic thickness represents a measure of diaphragm size obtained in the zone of opposition [36]. Technically speaking, a B-mode ultrasound is used with a 7.5–10 MHz linear probe placed in parallel between the 8–10 intercostal spaces in the mid-axillary line in the area of opposition, either during tidal breathing or a maximal inspiratory effort, to assess diaphragmatic thickness [36]. The structures encountered when using an ultrasound beam include skin and soft tissues, intercostal muscles (hypoechogenic), parietal and visceral pleurae (hyperechogenic), the diaphragm (hypoechogenic), and parietal and visceral peritoneum (hyperechogenic) (Figure 2). A normal thickness measured at end-expiration in healthy volunteers is around 1.1–1.4 mm in women and 1.3–1.9 mm in men [43,44,45]. In general, a value of 1.73–2.19 mm at end-expiration is considered normal [46,47]. In invasively and mechanically ventilated patients, diaphragmatic ultrasound was recently validated, showing a high reproducibility of right hemidiaphragm thickness, with a good correlation with diaphragm electrical activity [41]. Thickness of the diaphragm is a reliable measure of weaning from the ventilator, showing a similar performance to other weaning indexes [48,49].

##### Diaphragmatic Thickening

Diaphragmatic thickening represents the contraction of the muscle during breathing to quantify the magnitude of the respiratory effort [36]. Diaphragm excursion is usually assessed both via B- and M-mode ultrasonography. An advantage of M-mode is that it visualizes the movement of the diaphragm over time and provides an accurate measurement of diaphragmatic displacement over a respiratory cycle [36]. Using the B-mode setting with a higher-frequency (>10 MHz) linear probe, the thickening fraction is calculated as a fraction of the difference between thickness at end inspiration and end expiration: (end-inspiratory thickness − end-expiratory thickness) divided by the end-expiratory thickness × 100 [36]. Excursion positively correlates with lung inspiratory volumes and is higher during forced inspiratory breathing [50,51]. The diaphragm appears thicker in the upright position compared to the supine position [52]. Several definitions and thresholds are available, making univocal interpretation difficult. Abnormal thickening has been defined as a fraction of less than 20% or a tidal excursion of less than 10 mm [53]. The percentage of thickening during normal breathing is around 30% and 35% on both sides in healthy men and women [44,54], while around 11% in mechanically ventilated subjects [41]. Some authors have reported a mean thickening fraction of 20% during tidal breathing, without significant difference between right or left hemidiaphragm in mechanically ventilated patients. However, the assessment of bilateral measures is difficult and poorly reproducible, and the right diaphragm is easier to investigate [41]. Another important variable is the ratio of the thickness between the two sides of the diaphragm. A normal ratio is set between 0.7 and 1.5 in healthy men and 0.6 and 1.6 in healthy women; moreover, values far from the threshold are representative of an imbalance between the two hemidiaphragms [44]. A meta-analysis in 1071 patients found that the diaphragm thickening fraction is highly specific and that diaphragmatic excursion is highly sensitive to the weaning outcome, with possible variability across different ICU populations [55]. Diaphragmatic ultrasonography demonstrated good sensitivity and specificity in predicting reintubation within 48 h from weaning [56]. Another meta-analysis and additional recent studies have confirmed diaphragmatic dysfunction as a predictor of weaning outcome [57,58]. The cutoffs more associated with weaning failure were 11–14 mm in excursion and 30–36% in thickening [48,59,60,61]. Chien et al. suggested the combination of a diaphragmatic ultrasound and echocardiography to assess weaning prediction [62]. This was also confirmed by Silva et al. in patients undergoing a spontaneous breathing trial in pressure support mode [63] and by Haji et al. in their T-tube trial [64]. Another approach is the combination of the rapid shallow breathing index (RSBI) with ultrasonography. This approach demonstrated that the RSBI seems to be more accurate in the prediction of successful weaning when used alone [65]. According to the available evidence, the use of diaphragmatic ultrasounds seems promising in the weaning phase and to diagnose diaphragmatic dysfunction. Despite this, the limitations of an ultrasound should be always considered, including the need for training, intra- and inter-operator variability, and patient characteristics.

##### Diaphragmatic Displacement

Diaphragmatic displacement is useful for investigating the cyclic caudal movement of the diaphragm during the respiratory cycle and can be visualized through B- and M-mode ultrasonography using a convex probe with a frequency of 3.5–5 MHz (Figure 3). This technique allows one to assess diaphragm displacement, the speed of contraction, and inspiratory and total respiratory cycle timings. A diaphragm excursion of 49 mm is considered normal in spontaneously breathing subjects [66]. During non-invasive ventilation, diaphragmatic excursion was recently considered to be a potential predictor of NIV response, showing a good sensitivity and specificity for a value of 1.37 cm [67]. In patients with severe COVID-19, diaphragmatic excursion at hospital admission can accurately predict the need for ventilatory support and mortality [68].

##### Intercostal Muscles Thickness

The external intercostal muscles extend from the tubercles of the ribs dorsally to the costochondral junctions ventrally, with fibers oriented obliquely. External intercostal muscles are activated during inspiration, whereas internal intercostal muscles are activated during the expiratory phase. De Troyer et al. showed that the third dorsal external intercostal is usually activated in the early phase of inspiration [69], and that the amount of the activation of the external intercostal muscles during basal breathing was associated with the degree of their mechanical response [70]. Among external intercostal muscles, the parasternal seem to be those with more inspiratory action on the lungs despite a having lower pressure-generating ability than the other external intercostal muscles [71]. Interestingly, Sampson et al. demonstrated that chest wall deformation during breathing depends on the coordination of inspiratory intercostal muscles, being parasternal and not necessarily the most prominent [72]. In summary, during breathing, external intercostal muscles contract, generating torque more generated toward the lower rib than the upper rib, thus raising the ribs as a final effect. On the contrary, internal intercostals muscles contract, generating torque to the lower ribs [73].

The evaluation of intercostal muscles with ultrasound has been recently introduced. The parasternal intercostal muscles can be investigated using a 10–15 MHz, linear transducer in M-mode, placed 3–5 cm laterally to the sternum, in the sagittal plane, between the second and the third rib [73]. The patient is placed 20 °C head up and the pleural line is observed in B-mode as a “bat sign”, just above the parasternal intercostal muscle with the three biconcave layers, as well as the two linear hyperechoic membranes from the anterior and posterior ribs. Thickness is measured between the fascial borders as hyperechogenic structures. During inspiration, increased thickness is observed, and the rib cage is moved cranially and anteriorly. Thickening fraction is calculated as follow = [(End insp thickness − End exp thickness)/End exp thickness × 100] [73]. In a human setting, whether the usefulness of diaphragmatic ultrasounds for measuring is questioned or not, the literature on intercostal muscles is scarce. Nakanishi et al. showed that both the diaphragm and intercostal muscles become atrophic and lose thickness following excessive inspiratory support [74]. Dres et al. showed that the parasternal intercostal muscle-thickening fraction was associated with a failed spontaneous breathing trial in patients who were mechanically ventilated. Patients with diaphragmatic dysfunction showed a fraction exceeding 8%, while 10% was predictive of weaning failure [75]. Umbrello et al. reported that patients without diaphragmatic dysfunction showed a higher diaphragm (>30%) and lower parasternal intercostal thickening fraction (<5%) compared to patients with diaphragm dysfunction [76]. Yoshida et al. proposed the evaluation of intercostal muscle thickness via ultrasound at rest and during maximal breathing, showing a significant increase in the thickness of the intercostal muscle in the first [from mean (standard deviation) of 1.97 (0.66) to 2.51 (0.94)], second [from 2.17 (0.83) to 2.62 (0.95)], third [from 2.65 (1.13) to 3.19 (1.24)], fourth [from 2.79 (0.86) to 3.55 (1.02)], and sixth [from 2.52 (0.52) to 2.80 (0.66)] intercostal spaces of the anterior portions [38]. The usefulness of assessing the ultrasound thickness of intercostal muscles is still to be determined. However, evidence is encouraging, suggesting that a low parasternal intercostal thickening fraction may reflect low inspiratory effort, whereas low or high levels may reflect elevated inspiratory work by extra-diaphragmatic muscles, depending on the mechanical respiratory support provided by the ventilator [73].

#### 1.3.2. Limb Muscles

The rectus femoris is the most commonly investigated muscle, likely because it is easy to identify and analyze with a single image, and because it is considered a functionally important muscle for the performance of daily living and, at the same time, is subject to significant wasting during bedrest and illness, more than muscles of the upper limbs [77]. This technique allows for the assessment of both muscle mass (thickness or cross-sectional area—CSA) and quality (echodensity), as well as an estimation of the muscle’s force-generating capacity (the pennation angle).

Rectus femoris ultrasound is generally performed using a high-frequency linear transducer array probe (8–12 MHz), using the B-mode setting. Briefly, patients are studied in the semirecumbent position with extended knees; the probe is placed on the anterior part of the thigh, at 2/3 of an imaginary line connecting the anterior superior iliac spine and the midpoint of the proximal border of the patella. A mark can be drawn on the skin to increase the reproducibility of the subsequent measurements. The transducer is oriented transverse to the longitudinal axis of the thigh at a 90° angle; the probe is coated with water-soluble transmission gel to increase the acoustic contact, and care is taken to reduce the pressure on the tissues and the consequent distortion of the image as much as possible. Typical values of quadricep thickness and rectus femoris CSA in healthy volunteers have been reported to be 2.6 cm [78] and between 4.53 and 8.68 cm^2^ [78,79,80], respectively. On the other side, in critically ill patients, average values at ICU admission have ranged between 0.98 and 2.23 cm for quadricep thickness [81,82,83] and from 2.26 to 4.42 cm^2^ for rectus femoris CSA [5,82,83,84,85].

Information about muscle composition can be gathered using the quantification of muscle echodensity, which is calculated by performing a grey-scale analysis of image pixels, using standard software for image editing. This process has been shown to correlate with bioptic findings [86] as it reflects the muscle composition: an increased echogenicity represents a more homogenous muscle [86]. Quantification of muscle echodensity requires exporting the muscle ultrasound scan as a digital image file for subsequent, offline computer analysis, and the absolute value of density of the image critically depends on the settings with which the image was acquired with.

Changes in quadricep muscles’ echodensity have been associated with negative outcomes [80]. Eventually, muscle architecture can be described using the pennation angle, i.e., the angle of the insertion of muscle fibers into their aponeurosis, which provides information about muscle strength: the larger the pennation angle, the more contractile material is present, and thus the higher the capacity to produce force [87,88]. The rectus femoris pennation angle is measured using the same method and in the same position of muscle area and thickness; a longitudinal view is obtained by rotating the probe parallel to either the lateral or medial head of the muscle. Few studies have investigated the pennation angle in critically ill subjects; in healthy subjects, the average pennation angle of the rectus femoris has been reported to range from 8.76 ± 1.78 [77] to 17.5 ± 3.9° [89]. In critically ill patients at ICU admission, the pennation angle was 10.8 ± 2.6° [84], and an angle <4.4° was found to be associated with a worse outcome [90] (Figure 4).

It is well known how limb muscle size, structure, and function deteriorate during the course of ICU stay, i.e., by approximately 3% per day in the first week of ICU stay [91]; sonographic findings of reduced rectus femoris CSA have been found to be associated with poor clinical outcomes [92]. Using a CT scan as the reference method to define low muscle mass, a cutoff value of <2 cm for the thickness of the quadriceps’ muscle layer thickness and a rectus femoris CSA of <4.7 cm^2^ had an AUC of 0.84 and 0.76, respectively [93]. However, several limitations have to be carefully considered when using specific cutoffs for muscle US, specifically, the lack of external validation in the majority of the studies, and the lack of standardization for the site of measurements.

The use of US to measure muscle mass has consistently been found to be reliable in several investigations. Puthucheary et al. [91] reported an excellent coefficient of determination (R2) of 0.97 for the measurements of rectus femoris CSA using two blinded independent raters. Grimm et al. [94] evaluated the reproducibility of muscle echogenicity, showing excellent inter- (0.915) and intra-rater (0.972) coefficients. The ultrasound measurement of the rectus femoris size was recently validated through a comparison with a CT-derived skeletal muscle area at L3 level in a prospective observational trial on 200 non-critically ill patients; a combined score of the ultrasound measurement, together with sex, height, and weight, predicted muscle mass with an R2 of 0.74 [95]. In 15 critically ill subjects, the interrater reliability of the rectus femoris CSA had an intra-class correlation coefficient between 0.87 and 0.9; feasibility, defined as the percentage of measurements that were obtainable, ranged from 75% to 100% [96]. A study of muscle US image acquisition by physical therapists and students found an average intraclass correlation coefficient for all rates of 0.903, indicating excellent reliability of image acquisition regardless of the level of experience of the operator, severity of patient illness, or patient setting [83]. Another longitudinal, validation study assessed the intra- and inter-observer reliability of muscle ultrasound in 29 long-stayer, critically ill subjects; the authors evaluated two measurement sites: at the midpoint or at two-thirds of the length between the anterior superior iliac spine and the upper border of the patella. Intra- and inter-observer reliability ICC scores were 0.74 and 0.76 at the “midpoint” and 0.83 and 0.81 at the “two-thirds” site, respectively, showing that the method is reproducible, with a higher reliability at the two-third site [97].

Previous studies showed a negative correlation between muscle size and quality and ICU length of stay or mortality [5,92,97,98]. More recent studies have confirmed these findings. For example, a 3-week follow-up analysis of rectus femoris CSA in ICU trauma patients showed that 100% of participants experienced severe muscle mass loss, and 45% of rectus femoris muscle mass was lost by day 20, together with a progressive increase in echogenicity score [99]. A recent single-center observational study in 35 young trauma patients staying in the ICU for at least 7 days found that the rectus femoris cross-sectional area, mid-arm circumference, and calf circumference were reduced rapidly during the first week of the ICU stay, whereas the relationship between muscle loss and the clinical outcome was less defined [100].

With the aim of investigating the degree of change in rectus femoris muscle size over time, Wu et al. prospectively enrolled 284 critically ill subjects; the authors confirmed an average daily muscle atrophy rate of about 1%, with the highest reduction occurring in the third and fourth weeks of stay; daily atrophy rates were approximately three-times higher in women than in men, and protective factors of muscle atrophy included higher BMI and lower initial muscle size [101]. An observational study on 74 critically ill subjects analyzed the time course of rectus femoris muscle thickness on the first, third, and seventh days of ICU stay; the muscle size was reduced by 15% on average over the first week, and a greater reduction was associated with worse clinical outcomes [102]. The association between US-assessed muscle mass and muscle strength generation was evaluated in an observational investigation on 37 septic patients: the authors found a significant association between a decrease in rectus femoris CSA between the second day of stay and ICU discharge and a lower handgrip strength at hospital discharge, suggesting the significant clinical impact of muscle wasting [103]. A recent longitudinal investigation confirmed how severe and critical COVID-19 patients showed a 30% reduction in the rectus femoris cross-sectional area, with an average 16.8% increase in echodensity from days 1 to 10 [104]. A strong correlation between increased echodensity and inflammation was confirmed in several muscle biopsy studies [3].

Several recent investigations have focused on the factors associated with muscle wasting: Lee et al. conducted a prospective observational study to determine the association between baseline quadricep muscle status, premorbid functional status, and 60-day mortality in 90 patients. The authors found that every 1% loss in quadricep muscle thickness over the first week of critical illness was associated with 5% higher odds of 60-day mortality; moreover, a higher nutrition risk, sarcopenia, and frailty at baseline was associated with lower baseline muscle size and higher 60-day mortality, suggesting a complex relationship between premorbid functional status, muscle mass, and outcome [105]. Mukhopadhyay et al. demonstrated that a higher nutritional risk assessment at admission is associated with a higher subsequent muscle reduction, allowing them to identify patients at risk of muscle loss [2]. A recent observational study in a sample of critically ill, COVID-19 patients found an average 30% reduction in rectus femoris CSA over the first week of ICU stay, with a significantly higher reduction in non-survivors, together with a significant increase in muscle echodensity over the first week, again of a higher extent in non-survivors. Interestingly, the change in rectus femoris area was related to the cumulative protein deficit over the first week of ICU stay, suggesting that changes in muscle size and quality are related to the outcome of critically ill patients and are influenced by nutritional management strategies [106]. In a longitudinal investigation, Yanagi et al. found that, among 72 critically ill patients staying in the ICU for >2 days, low quadricep muscle mass at ICU discharge was associated with a low muscle function, as assessed using the Medical Research Council sum score, and was associated with an almost four-times higher 1-year mortality, highlighting the utility of muscle mass measurements to identify high-risk patients and suggesting the use of muscle mass as a relevant patient-centered outcome [107]. Muscle ultrasound has also been shown to be an early predictor of physical disability: in a prospective investigation of 41 critically ill patients admitted to the ICU for respiratory failure or sepsis, the change in rectus femoris CSA over the first week of ICU stay was a strong predictor of muscle weakness at hospital discharge [80]. In another investigation, a loss in pennation angle was observed during the first week of ICU stay, and such loss predicted the subsequent development of ICU-acquired weakness in a sample of 50 critically ill and mechanically ventilated subjects [72].

A potential advantage of muscle ultrasound, as compared with other methods for the assessment of lean body mass, such as bioelectrical impedance, is that it can be less dependent on body hydration state. To test this hypothesis, da Silva Passos et al. recently published the results of a prospective cohort study. The authors compared the findings of a BIA-derived phase angle with rectus femoris CSA and found that only rectus femoris CSA was a significant predictor of mortality in a sample of 160 mechanically ventilated and critically ill subjects [82,90]. The authors described the relationship between rectus femoris CSA and quadricep muscle thickness, with volitional measures of strength and function at 7 days after the ICU admission of 29 patients with sepsis. The authors showed an expected decrease in both in rectus femoris CSA and thickness (23.2% and 17.9%, respectively), while only the rate of change per day of CSA was correlated with muscle strength on day 7. Similarly, Puthucheary et al. showed that thickness measurements significantly underestimate ICU muscle wasting compared with rectus femoris CSA [4].

In summary, the literature seems consistent in the utility of rectus femoris ultrasound (especially in the cross-sectional area at the lower third of the thigh) to track the loss of muscle mass and possibly muscle function and as a marker of the severity of an illness and possible negative outcomes. However, so far, no unanimous cutoffs to define sarcopenia or an increased risk of mortality have convincingly been reported, and the relationship of muscle ultrasound with nutritional or physical intervention is yet to be demonstrated. Part of this uncertainty strictly depends on the variability in probe settings, and a shared, standardized protocol is needed.

### 1.4. Electromiography

The mainstay of the prevention of ICUAW is the minimization of the risk factors assessed in prospective studies that unfortunately assessed subsets of ICU patients (severe sepsis, multiorgan failure, prolonged mechanical ventilation (greater than ∼7 days), or those receiving high doses of corticosteroids). When a group of patients with severe sepsis, multiorgan failure, or prolonged mechanical ventilation have undergone full electrophysiological and histological investigation, CIP and CIM have been found to significantly overlap [7].

Although the clinical assessment of muscle weakness using the Medical Research Council (MRC) score can quantify strength impairment, electromyography (EMG) remains the hallmark in diagnosing and differentiating ICUAW types [108].

Li et al. assessed peripheral nerve biopsies from the sural nerve from ICU patients and found a reduction in the sodium channel subtype Nav1.6 on the sural nerve [109]. This may potentially explain why its dysfunction affects neurological functions across all systems of the body during critical illness [109].

In a critical care setting, the inability to wean from ventilator support is common a symptom of the ICUAW method, and diaphragmatic EMG remains technically challenging within the ICU setting. An examination of the phrenic nerve (prolonged latencies or decreased motor unit action potentials) and diaphragm EMG (pattern of fibrillations and positive sharp waves or reduced number of motor unit potentials) can assist with diagnosis [110].

However, several factors can limit an EMG examination’s effectiveness and influence the results. A lack of patient cooperation and interactions with other electronic devices may obscure both nerve conduction and EMG results, providing technically inadequate studies for diagnosis. In addition, very often, patients presenting with anasarca and hypothermia can alter the amplitude and velocity of recordings in nerve conduction studies. The gold standard for the diagnosis of critical illness neuropathy remains electrodiagnostic testing, which includes nerve conduction studies and needle electromyography [111]. The evaluation of profound weakness in the ICU setting should be implemented, and electrodiagnostic testing is an essential tool that can direct the clinical team in determining further management [111].

## 2. Nutritional Outcome and Metabolic Assessment

Acute critical illnesses can determine the significant deterioration of one or more vital functions with profound biological, biochemical, metabolic, and functional modifications. The entity of this stress response mainly depends on the severity of the clinical state and its duration. Whether the cause of critical illness is trauma, septic or surgical stress, or a patient’s nutritional status and nutritional risk could guide metabolic support [112]. Nutritional status upon admission remains complex, and it is not clear whether the long-term physical limitations after critical care are attributable to the impairments acquired during ICU stay, or to preadmission functional impairment caused by chronic disease or general frailty [11]. It might be difficult to provide appropriate nutritional support to critically ill patients due to prolonged fasting periods, delivery hurdles brought on by insulin resistance, and gastrointestinal dysfunction. There is not much data to support nutrition advice, especially when it comes from outcomes such as muscle mass, strength, and function. The loss of LBM is a clear marker of malnutrition. However, only a few RCTs of nutrition therapies in critical illness included an endpoint of muscle mass, strength, or function [113,114,115,116,117,118,119]. Despite increasing evidence, there is still no study which has confirmed that nutrition interventions are useful in enhancing any muscle strength or function in critically ill patients. Therefore, the use of LBM to guide nutritional assessment and therapy in critical illnesses is still highly debated.

## 3. Future Directions and Implications

An innovative tool to potentially assess muscle mass is the use of near-infrared spectroscopy (NIRS) technology. In the near-infrared spectrum (700–1100 nm), photons are capable of several centimetres of tissue penetration, before being absorbed by metalloproteinases such as hemoglobin, myoglobin, and mitochondrial cytochrome oxidase. Briefly, NIRS yields values of tissue oxygen saturation of hemoglobin (StO2), which represents spatially integrated information from arterioles, capillaries, and venules, and provides information about oxidative metabolism and the intramuscular matching between O2 delivery and utilization, that can provide the link between catabolism, inactivity, and the loss of lean body mass. In a proof-of-concept physiological investigation involving 26 healthy participants, NIRS was found to be a reliable tool to investigate skeletal muscle oxidative capacity [120]. In healthy subjects, even a short (10-day) period of horizontal bed rest was shown to impair in vivo oxidative function during exercise, as assessed using NIRS, and muscle catabolic processes induced by inactivity were demonstrated to be less energy consuming than anabolic ones [121]. Since the penetrability of light into tissues is proportional to its wavelength, NIRS has also been suggested to be able to assess the depth of subcutaneous tissue. In 93 healthy subjects, NIRS was found to be significantly associated with lean body mass [122]. Similar findings were reported for non-critically ill and hospitalized patients, using dual-energy X-ray absorptiometry as the comparator for lean body mass [123]. For the assessment of metabolic bone illnesses such osteoporosis, sarcopenia, and obesity, dual-energy X-ray absorptiometry is currently considered as one of the most adaptable imaging modalities. However, in a critical care setting, the use of this tool is still infrequent, despite several reports in non-critically ill patients [11,124] revealing its utility in detecting changes in muscle mass. Reference standard values at X-ray absorptiometry in all age categories were investigated by Imboden et al., who found that men had a higher mean lean mass than women (60.8 kg vs. 42.3 kg). From the youngest to oldest age categories, the mean lean mass significantly declined for both men and women. Men’s mean lean mass was shown to fall with increased age, but women’s mean lean mass remained stable until the fifth decade, at which point it began to decline [125].

Potential innovations in the bedside assessment of muscle mass by means of ultrasonography include shear wave elastography, superb microvascular imaging, and contrast-enhanced ultrasound. The first is a technique used to measure tissue stiffness because of a disease, which is based on the generation of shear waves determined by the displacement of tissues induced by the force of a focused ultrasound beam; superb microvascular imaging is an innovative technique designed for imaging microvascularization that cannot detect using a color Doppler, allowing the visualization of low-velocity and microvascular flow. The latter involves the administration of intravenous contrast agents such as gas-filled microbubbles to better visualize organs and blood vessels. A recent observational investigation compared the findings of these techniques between critically ill patients diagnosed with ICU-acquired weakness and healthy controls and identified specific features of the muscle of critically ill subjects, shedding a light on these innovative imaging modalities [126]. A recent validation study investigated the reliability and reproducibility of shear-wave elastography muscle measurements in critically ill patients; the authors found how inter-operator reproducibility and intra-operator reliability were well above 0.9 [127]. In 130 non-critically ill patients, lower leg shear-wave velocities were compared between sarcopenic and non-sarcopenic subjects: velocities in the sarcopenia group were significantly smaller than in the healthy control group and were positively correlated with an appendicular skeletal muscle mass index and grip strength [128]. Recently, the measurement of temporalis muscle thickness was proposed as a measure of muscle wasting in neuro-critically ill patients. In an observational trial, a decrease in rectus femoris size, as assessed via ultrasound, was paralleled by the ultrasound measurements of temporalis muscle thinning; moreover, the finding was also confirmed by CT-based measurements of the temporalis muscle size and lay the foundation for the assessment of the temporal changes of other muscle groups in critically ill subjects [129].

The measuring and tracking of LBM have important implications: LBM assessment may provide crucial chances to identify critically ill patients at high nutritional risk at an early stage and to direct and evaluate metabolic care after ICU admission. Bioelectrical impedance analysis, musculoskeletal ultrasound, and CT-scan analysis are becoming popular in ICU [130]. The strengths and limitations of these various methodologies must be taken into consideration when interpreting the results. Exciting new research in this field has focused on the quantity as well as the quality of lean body mass, thus providing information on the infiltration of adipose tissue and intramuscular glycogen storage. Lean body mass measurement techniques are continuously being improved to increase their usefulness at the bedside, providing useful tools to clinicians to direct metabolic assistance [130].

## 4. Conclusions

Mechanical ventilation and critical illnesses predispose reduced muscles’ strength, prolonging ICU duration and complications. Several tools (including computed tomography, bioelectrical impendence, ultrasound, electromyography, near-infrared spectroscopy, and dual-energy X-ray absorptiometry) are currently being used to assess lean body mass; however, their use in critical illnesses is still poor. Future research is warranted to better address the utility of these tools in critical care.

## Figures and Tables

**Figure 1 diagnostics-13-00888-f001:**
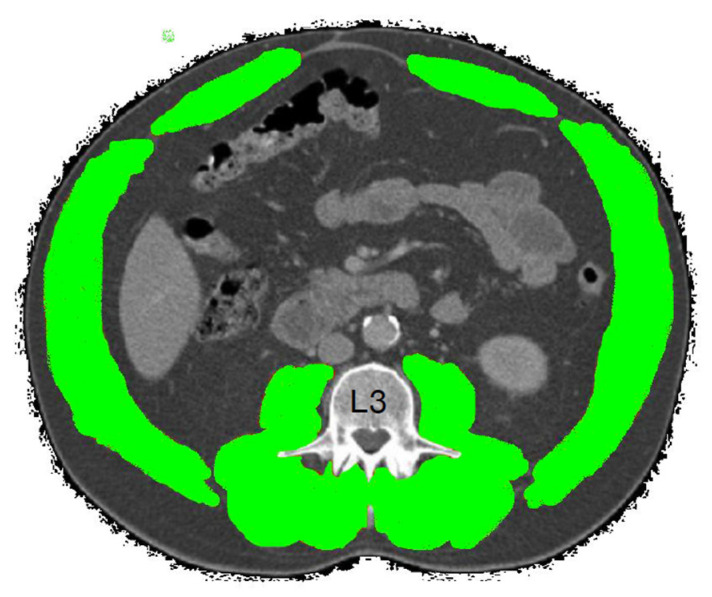
A cross-sectional computed tomography (CT) image at the third lumbar vertebra (L3) showing skeletal muscle segments. Skeletal muscle segmented in green color.

**Figure 2 diagnostics-13-00888-f002:**
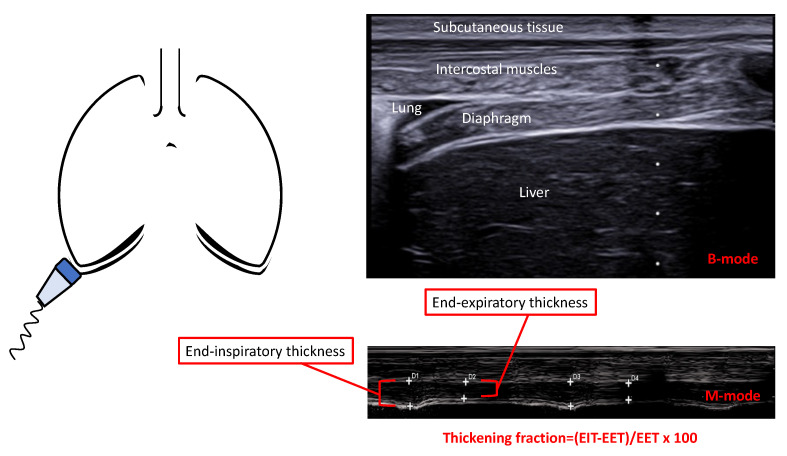
Ultrasonographic assessment of diaphragm thickness and thickening. Ultrasonographic assessment of diaphragmatic thickness visualizing the normal diaphragm in the zone of opposition using a 7.5–10 MHz linear probe. View in B-mode and M-mode of the diaphragmatic thickness. In both modalities, the probe is placed parallel to an intercostal space between the 8th and the 10th spaces. Diaphragmatic thickness is measured at end inspiration and end expiration, and the thickening fraction is calculated according to the formula = (EIT − EET)/EET × 100.

**Figure 3 diagnostics-13-00888-f003:**
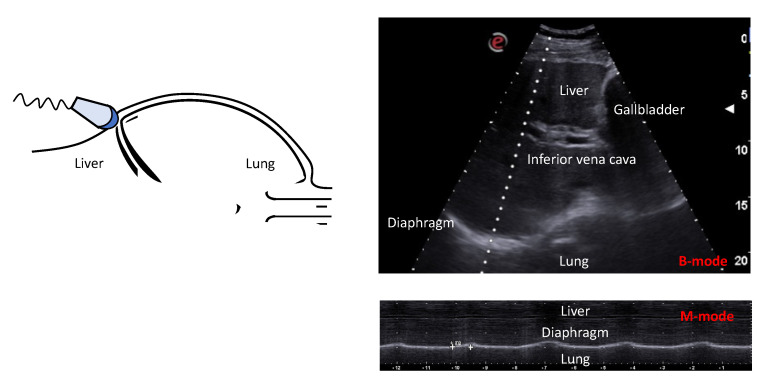
Ultrasonographic assessment of diaphragm displacement during inspiration and expiration. Ultrasonographic view of the normal diaphragm in the region of the liver dome in B- and M-modes during inspiration and expiration. This technique uses a 3.5–5 MHz convex probe placed between the midclavicular and anterior axillary lines, directed cranially, medially, and dorsally in the region of the liver dome. This technique allows one to measure diaphragm displacement, contraction speed, inspiratory time, and total respiratory timing.

**Figure 4 diagnostics-13-00888-f004:**
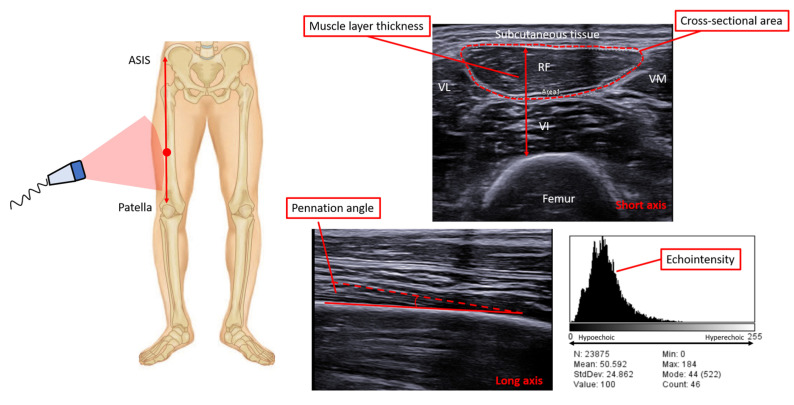
Ultrasonographic assessment of the quadriceps muscle. **Left** panel: the image shows the standardized level of the ultrasound scan of the lower limb; in the supine position, the linear probe is placed on the anterior part of the thigh, at 2/3 of an imaginary line connecting the anterior superior iliac spine (ASIS) and the midpoint of the proximal border of the patella, with the probe perpendicular to the muscle. **Upper right** panel: the figure shows a transverse scan of the quadricep muscle, which is composed of three vastus muscles (medialis, intermedius, and lateralis) and the rectus femoris; the red dashed line represents the rectus femoris cross-sectional area, the double-arrow line depicts the quadriceps muscle layer thickness. **Lower middle** panel: the picture shows a longitudinal scan of the quadricep muscle; the pennation angle is measured at the intercept of the fascicular path (dashed line) to the lower aponeurosis (solid) line. **Lower right** panel: the diagram shows the grayscale histogram in a transverse axis of the rectus femoris. VI vastus intermedius, VM vastus medialis, VL vastus lateralis.

## Data Availability

Not applicable.

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
