# Peer review of "Update on Lean Body Mass Diagnostic Assessment in Critical Illness"

_diagnostics, 2023, doi:10.3390/diagnostics13050888_

Round 1
Reviewer 1 Report
I read your review “Update on lean body mass diagnostic assessment in critical illness” with enthusiasm and interest. Lean body mass is an important factor affecting prognosis in critically ill patients. A nice attempt that is relevant and pertinent in today’s context however there are a few suggestions for the authors
1. Page 1 line 22- “A lack of uniform bedside management tools could impact the nutrition outcome.” Is it management or measurement tool?
2. Why dual-energy x-ray absorptiometry and its role on lean body mass is not explored?
3. Introduction is not coherent.
4. Nutrition outcome and Metabolic assessment should have been elaborated on in the review.
Author Response
Submission to: Diagnostics – Special Issue: Critical Care Imaging
Manuscript title: Update on lean body mass diagnostic assessment in critical illness
Manuscript classification: Review
Editor-in-Chief,
Prof. Dr. Andreas Kjaer
We would be very grateful if you could consider our revised manuscript “Update on lean body mass diagnostic assessment in critical illness” for publication on Diagnostics. The manuscript has been carefully revised according to Reviewer’s comments and suggestions. All changes have been highlighted in red color in the revised submission. We have also provided a point-by-point response to the reviewers’ comments. We hope that the present manuscript can be now accepted for publication.
We declare:
1) All authors have made substantial contributions.
2) All authors have agreed to conditions noted on the Authorship Agreement Form.
3) The author Denise Battaglini is Guest Editor of the submitting Special Issue – Critical Care Imaging.
4) No overlap with previous publications is present; the manuscript, including related data, figures and tables is not under consideration elsewhere.
5) The principal authors take full responsibility for the manuscript, the analyses and interpretation of literature, and the conduct of the manuscript; has full access to all of the data; and has the right to publish any and all data separate and apart from any sponsor.
6) Indication that the Author has received patient consent form for any figure or video of any recognizable participant: not applicable.
7) The Authors declare that they received an invited free (100%) voucher to publish on Diagnostics – Special Issue Critical Care Imaging.
We believe that our manuscript is compliant with journal’s requirements.
Sincerely
Silvia De Rosa and Denise Battaglini on behalf of co-authors.
Reviewer 1
I read your review “Update on lean body mass diagnostic assessment in critical illness” with enthusiasm and interest. Lean body mass is an important factor affecting prognosis in critically ill patients. A nice attempt that is relevant and pertinent in today’s context however there are a few suggestions for the authors.
Response: Thank you very much for appreciating our work.
Page 1 line 22- “A lack of uniform bedside management tools could impact the nutrition outcome.” Is it management or measurement tool?
Response: Thank you very much for this comment. We have modified this typo accordingly (red color).
Why dual-energy x-ray absorptiometry and its role on lean body mass is not explored?
Response: Thank you very much for this comment. A brief paragraph on the utility of dual energy X-ray absorptiometry has been added in the section “Future directions”.
Introduction is not coherent.
Response: Thank you very much for this comment. We have modified the introduction to make it more coherent with the content of this manuscript.
Nutrition outcome and Metabolic assessment should have been elaborated on in the review.
Response: Thank you very much for this comment. We have added a paragraph on Nutrition outcome and metabolic assessment as suggested.
Reviewer 2
Comments and Suggestions for Authors
Dear Authors,
Thank you for a very nice and interesting piece on body composition measurements.
Response: Thank you very much for appreciating our work.
It would be interesting to add a section on implications. What would the benefits be for critical care patients if we were able to apply body composition measurements daily?
Response: Thank you very much for this comment. We have added this section to the “Future directions” re-named as “Future directions and Implications”

Reviewer 2 Report
Dear Authors,
Thank you for a very nice and interesting piece on body composition measurements.
It would be interesting to add a section on implications. What would the benefits be for critical care patients if we were able to apply body composition measurements daily?
Author Response

(The authors gave the same response as above.)
